# Design and Evaluation of Screening Smoke Compositions Based on Red Phosphorus in Open Field Conditions

**Bogdan Gheorghe Pulpea** [1], **Daniela Pulpea** [1,*], **Eugen Trană** [1,*], **Traian Rotariu** [1], **Raluca Elena Ginghină** [2], **Gabriela Toader** [1] and **Florin Marian Dirloman** [1]

[1] Military Technical Academy "Ferdinand I", 39–49 George Coșbuc Boulevard, 050141 Bucharest, Romania
[2] Research and Innovation Center for CBRN Defense and Ecology, 225 Șoseaua Oltenitei, 041327 Bucharest, Romania
*   Correspondence: pulpea.daniela@gmail.com (D.P.); eugen.trana@mta.ro (E.T.)

**Abstract:** This research describes the design and evaluation of screening smoke compositions based on red phosphorus (RP), in open field conditions. In defense applications, smoke is used for both signaling and screening. Defense forces use smoke screening over their operational areas to conceal positions and activities from the ground or air observation performed by the enemy. In this research, various optimized smoke charges based on RP were prepared and tested to investigate IR and VIS screening effectiveness in static and dynamic conditions and to establish the parameters that influence the screening time (ST- the active time of the small solid hot particles suspended in the air). In addition, this study projected a mathematical model to describe the concealing time of a civil or military target by optimizing the smoke compositions. The mathematical approach clarifies the limitations of reducing smoke charge while maintaining adequate screening time. The empirical mathematical model estimates the screening time of the generated solid smoke particles (aerosol) through laboratory experiments and open field studies. According to the experimental results, more hot particles should be kept in the atmosphere to maintain the smoke screen and sustain the aerosol density in the considered volume.

**Keywords:** screening time (ST); smoke-generating devices (SGD); red phosphorus (RP)

## 1. Introduction

Smoke-generating pyrotechnics are frequently used for various military applications, such as colored smoke for ground signaling, ground-to-air signaling and marking, and white smoke for concealing a target [1]. Instant invisibility produced by smoke-generated obscurant systems that temporarily block optoelectronic sensors can substantially increase military personnel's survivability in combat situations.

Obscurant systems are based on the dispersion of a pyrotechnic smoke charge and the generation of a smoke screen formed between the enemy and the military technique. For a specific amount of time, this conceals the target from optoelectronic observation and range-finder devices in the VIS and IR domains of the electromagnetic spectrum. The main observation and range-finder technologies for seeing through compromised optical environments are light detection and range (LiDAR) [2], radio detection and range (RADAR) [3], and passive sensors based on image intensifier and thermal view [4]. However, the most common surveillance technology used in military operations is the last category stated above.

Choosing the suitable IR-screening compositions involves a trade between hazard and performance characteristics. The most effective mixtures are usually toxic or incendiary, while practical and safe alternatives are frequently absent. Others are unstable in storage and require specialized equipment to use. White phosphorus is both the best-performing and the most dangerous mixture. This pyrophoric compound is flammable, poisonous, and can cause adverse effects to wetland areas when disseminated by a blasting device [5,6]. Several injuries

and deaths have been linked to pyrotechnic hexachloroethane compositions (HC), which generate a thick hygroscopic zinc chloride smoke mixed with soot and chlorinated organics and is also toxic [7,8]. When hygroscopic liquids such as chlorosulfonic acid or titanium tetrachloride are dispersed, they form acidic clouds that are corrosive [6]. Because hydrocarbon and glycol oils are highly volatile, they produce a very effective and generally benign smoke screen; however, this usually necessitates the installation of a smoke generator [9,10]. Although titanium powder is an excellent nontoxic visual obscurant, it is difficult to scatter effectively, and the resulting screen has a short lifespan [11].

Black phosphorus has been studied extensively lately for its (opto)electronics, transistors, catalysis, and biomedical properties [12]. Additionally, black phosphorus-containing smoke generating devices (SGD) are already patented [13]. As a result, producers may be interested in black phosphorus smoke compositions. Even though it has many advantages, the present synthesis methods are difficult and expensive to implement, and commercial sources are not readily available [14].

Emerging new obscurant compositions are composed of using black phosphorus analog nanomaterials [15], Te/Se heterostructure-based [16], and MXene-based nanostructures [17], but these are only in the research phase and are used in smaller, costly applications such as batteries and solar cells. To be utilized in smoke generating systems, they must be produced in large quantities so that an entire army may be equipped with obscuring devices.

Although red phosphorus compositions are less incendiary and dangerous than white phosphorus, they have substantial sensitivity and aging issues [6]; RP remains the only widely available choice for industrial production of IR smoke-generating devices (SGD) for military troop endowment [18]. In this work, a RP-based obscuring composition was employed for loading several SGD that will be subjected to testing to determine an empirical relationship of the screening time.

Since there is a growing interest in evaluating the smoke screen efficiency through practical methods, this research comprises optimized smoke compositions based on red phosphorus (RP), potassium nitrate ($KNO_3$) and an organic binder (iditol resin or paraffin), to investigate their physical, chemical, and thermal properties. Moreover, obscuring characteristics through laboratory and open field fire testing were practically examined. Based on the experimental results, a theoretical determination of the IR screening time (ST) of an SGD can be predicted. Although an effective method to evaluate ST was also addressed by Klusáček [19] by firing and comparing different types of composition and a numerical approach was proposed by Chen-Guang [20] by applying the diffusion equation, herein, an upgraded, practical and straightforward hypothesis is stated, according to which ST is inversely correlated to the observation probability. This parameter was calculated based on Johnson Criteria that associates the smoke transmittance and the number of cycles [21] that the system must resolve across the target to perform a target detection or discrimination task with 50% probability [4]. Therefore, we also anticipated a ST mathematical approach as a function of smoke device characteristics, smoke dispersion and aerosol concentration together with meteorological conditions, target information and thermal sensor performance.

## 2. Theoretical Approach

The state-of-the-art in the field describes the characteristics of screening by determining the transmittance, which is a characteristic that just theoretically defines the quality of the smoke. In the case of a real military combat situation, smoke screening time is the only essential data because transmittance value fails to provide all of the required information that a soldier needs to decide, in real-time, the best procedure for obscuring a target. The concealing area and available standing time [20] have been mentioned as essential parameters to evaluate the smoke protection capability associated with a diffusion equation. Other researchers investigated the timing of ignition and burning [19] in relation to effective

screening and smoke cloud dissemination. However, no research has been found to date on IR screening time analysis and how we can ascertain this parameter.

The IR screening time (ST) is an essential characteristic of the smoke cloud and a mandatory condition for an obscuring system. ST represents the value expressed in seconds for the screening efficiency, which determines the tactical period during which a target can pass through the enemy firing range unnoticed.

According to Johnson, the mathematical expression of the observation probability (*OP*) is [21]:

$$OP = \frac{N_t}{N_{50}} = \frac{v(\lambda)}{v_t} \cdot \frac{1}{N_{50}} = \frac{d_c}{R} \cdot \frac{v(\lambda)}{N_{50}} = \frac{\sqrt{L_t \cdot h_t}}{R} \cdot \frac{v(\lambda)}{N_{50}} \tag{1}$$

where *OP* is the observation probability, $N_t$—number of cycles resolved across the target (cycles/mrad), $N_{50}$—number of cycles resolved across target critical dimension (cycles/mrad), $v(\lambda)$—spatial frequency (cycles/mrad), $v_t$—target spatial frequency, $d_c$—target feature size (m), *R*—path length (km), $h_t$—target height (m) and $L_t$—target length (m).

Assuming that the screening time is inversely proportional to the probability of observation described by Johnson, a series of parameters were defined that are useful to establish the screening time.

As a first step, it is essential to calculate the mass of aerosols ($m_a$) generated by a certain amount of pyrotechnic composition ($m_{pc}$):

$$m_a = m_{pc} \cdot k_c \tag{2}$$

where $k_c$ is the combustion coefficient (%) defined as a relationship between the mass of the residue ($m_r$), mass of released gases after combustion ($m_g$)—calculate in relation (4) and the initial mass ($m_{pc}$) of pyrotechnic composition:

$$k_c = \left(1 - \frac{m_r + m_g}{m_{pc}}\right) \cdot 100 \tag{3}$$

$$m_g = \frac{V_{sp} \cdot m_{pc}}{22.418} \cdot M_g \tag{4}$$

where 22.418 (L/mol) represent the volume occupied by 1 mol of gas at STP (standard temperature and pressure), $M_g$ is the molar mass of gas elements (g/mol), and $V_{sp}$ is the specific volume of the composition (L/kg) determined in practice.

Meanwhile, the gas volume ($V_g$) generated by a specific pyrotechnic composition mass is equal to:

$$V_g = V_{sp} \cdot m_{pc} \tag{5}$$

IR screening capacity is characterized by the coverage coefficient parameter ($k_{oc}$) of the dispersed volume of the aerosols that is calculated using the ratio between the gas volume, $V_g$ and the total screened volume, *V*. In the case of the chamber test, *V* is the volume of the chamber ($V_{ch}$ = 0.512 m³—that is kept constant) and for the real field test *V* is the disseminated smoke volume ($V_s$)—experimental determinate for each charge (Figure S1 from Supplementary Materials):

$$V_s = L_s \cdot h_s \cdot l_s \tag{6}$$

$$k_{oc} = \frac{V_g}{V} = \frac{V_{sp}}{V} \cdot m_{pc} \tag{7}$$

where $L_s$ is length of the smoke cloud, $l_s$ is the width of the smoke cloud, and $h_s$ is the height of the smoke cloud.

Furthermore, the aerosol density ($\rho_a$) is also stated as the following relation:

$$\rho_a = \frac{m_a}{V} = \frac{m_a}{V_g} \cdot k_{oc} = \frac{m_{pc}}{V_{sp} \cdot m_{pc}} \cdot k_c \cdot k_{oc} = \frac{k_c \cdot k_{oc}}{V_{sp}} \tag{8}$$

The concentration path length products of smoke (*CS*) are defined as a final step before outlining the screening time in which *CS* is the product between the above aerosol density ($\rho_a$), smoke cloud width ($l_s$) and the active smoke coefficient ($k_s$):

$$k_s = \frac{S_a}{S_t} \tag{9}$$

$$CS = \rho_a \cdot l_s \cdot k_s = \frac{k_c \cdot k_{oc} \cdot k_s}{V_{sp}} \cdot l_s \tag{10}$$

where $S_a$—cloud active surface and $S_t$—cloud total surface.

According to the above-mentioned equations, the mathematical relationship that defines the IR screening time is:

$$ST = \frac{1}{OP} \cdot N_{50} \cdot CS \cdot f_a(\lambda) = \frac{R}{d_c} \cdot \frac{1}{\nu(\lambda)} \cdot CS \cdot f_a(\lambda) \tag{11}$$

$$ST = \frac{R}{\sqrt{L_t \cdot h_t}} \cdot \frac{1}{\nu(\lambda)} \cdot \frac{k_c \cdot k_{oc} \cdot k_s}{V_{sp}} \cdot l_s \cdot f_a(\lambda)$$

where $f_a(\lambda)$ being an experimental value defined as a proportionality coefficient and here $N_{50}$ is a parameter specific for detection level, equal to 1 according to literature [4,21].

## 3. Experimental Section

### 3.1. Design and Development of Pyrotechnic Composites

3.1.1. Materials

Red phosphorus (P—Sigma Aldrich, St. Louis, MO, USA), potassium nitrate (KNO₃— Sigma Aldrich, St. Louis, MO, USA), magnesium powder (Mg—Sigma Aldrich, St. Louis, MO, USA), barium nitrate (Ba(NO₃)₂—Sigma Aldrich, St. Louis, MO, USA), barium peroxide (BaO₂—Sigma Aldrich, St. Louis, MO, USA), phenol-formaldehyde resin—Iditol (C₁₃H₁₂O₂— UM Sadu SA, Bumbești Jiu, Romania), granulate white paraffin wax (C₂₆H₁₂—indiaMART, online marketplace, Uttar Pradesh, India), gunpowder (KNO₃—75%, S—10%, C—15% from Scientific Research Center for CBRN Defense and Ecology, Bucharest, Romania, with particle size < 1 mm), delay composition MS-2 (UM Băbeni SA, Băbeni, Bucharest), electric igniters (MJG Electric Match—UM Băbeni SA, Băbeni, Romania), and RR4 ignition composition (magnesium-based ignition composition that also contains barium nitrate—Ba(NO₃)₂, barium peroxide—BaO₂ and iditol—UM Băbeni SA, Băbeni, Romania) were used.

3.1.2. Preparation of the Pyrotechnic Compositions

The smoke generating composites were prepared using three different formulations (PC1, PC2, PC3) and their compositions are specified in Table 1. The mixture was optimized in order to obtain rapid propagation of the charge canister. Knowing that phosphorus-based mixtures are very sensitive to friction and impact [22], a wet mixing process was used in all compositions, with the binder being mixed in a liquid state (50% iditol solution in ethanol for PC1 and melted paraffin for PC2 and PC3) with P to prevent accidents and safer ingredient manipulation. The KNO₃ used in all compositions was dried, grinded, and sieved using collecting fractions between 0.2–0.8 mm. PC1 was subsequently dried in powder form. PC2 and PC3 were compressed into cylindrical shapes (θ = 11 mm and h = 3 mm) followed by drying.

A magnesium-based ignition composition was applied on each tablet's surface to ensure reliable combustion. Additionally, to maximize the effect, different proportions of each composition (PC1—12%, PC2—47%, PC3—41%) were loaded into the smoke generating devices to achieve the volumetric efficiency required for the final charge. The total weight of the composition was 170 ± 5 g, while the volume was approximately 89.5 cm³ and the load density was 1.76 ± 0.2 g/cm³.

**Table 1.** Codes and composition of the primary components.

| Name | Components (wt%) | | | |
|---|---|---|---|---|
| | KNO$_3$ | P | C$_{13}$H$_{12}$O$_2$ | C$_{26}$H$_{12}$ |
| PC1 | 70 ± 0.5 | 28 ± 0.2 | 2 ± 0.5 | - |
| PC2 | 55 ± 0.5 | 35 ± 0.5 | - | 10 ± 1 |
| PC3 | 40 ± 0.5 | 50 ± 0.5 | - | 10 ± 1 |

### 3.1.3. SGD Manufacturing

The cartridge design (real representation in Figure S2) consists of a cardboard-sheet cylinder that, besides the RP smoke composition charge, contains other components that ensure the functioning of the SGD, such as: ignition mixture (1 g of black powder to ensure a throw distance of 25 ± 5 m with a velocity of 15 ± 2 m/s); delay composition (0.05–0.08 g MS2 that allows a standby time of 0.7–1 s until functioning); booster composition (0.8 g of black powder that provides the SGD body fragmentation, dispersion and ignition of the SGD charge). To ensure a safe functioning distance, an electric igniter was used to initiate the starter mixture. The RP load (Figure 1a), and the SGD assembly (Figure 1b) are represented below.

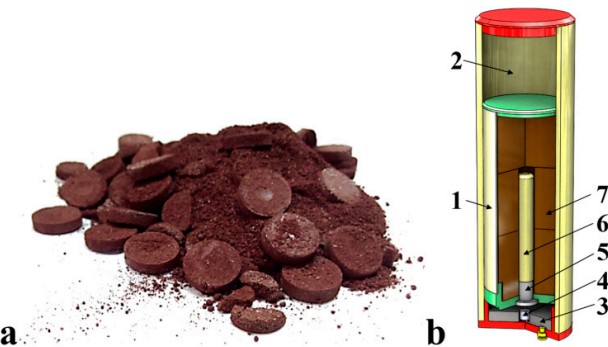

**Figure 1.** (**a**) RP smoke load and (**b**) SGD representation: 1—SGD body, 2—launcher tube, 3—gun powder, 4—electric igniter, 5—delay device, 6—central booster with gun powder, 7—RP smoke load.

Smoke screen formation has 3 steps as shown in Figure 2: firstly, PC1 was instantly combusted and formed the main dense smoke cloud; then, PC2 was burned and increased the surface of the smoke screen, and PC3 was ignited and reached the ground, supplying more smoke to the obscuring screen (more detailed representation is shown in Figure S3, from Supplementary Materials). The description of the SGD functioning mechanism, from fragmentation to smoke dissemination, has been previously studied [23,24] and represents the optimal model for screening targets used in a real open field.

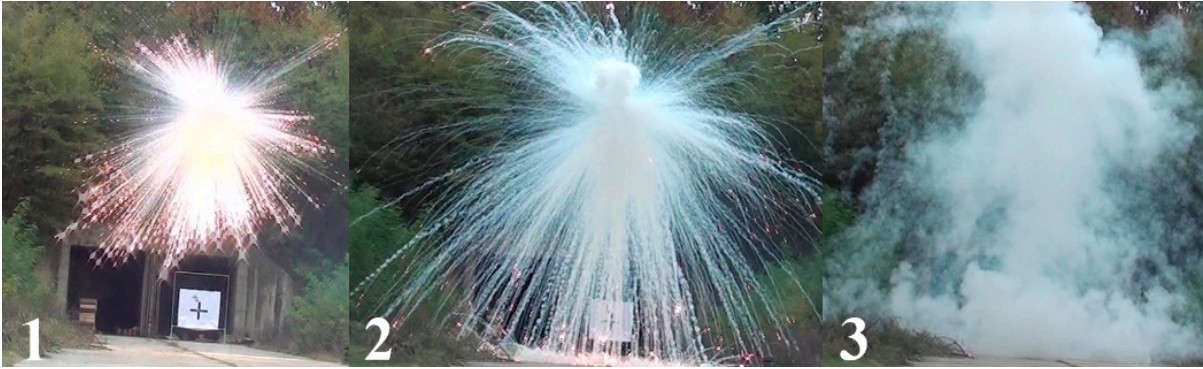

**Figure 2.** Three-step smoke screen formation: 1—PC1 combustion, instant smoke screen; 2—PC2 and PC3 combustion, smoke screen formation; 3—PC3 continue to supply the smoke screen.

### 3.2. Characterization

The thermal behavior of the smoke compositions was analyzed through differential thermal analysis (DTA)—OZM 551-Ex System—Czech Republic (Meavy software 2.2.5.43), to evaluate the influence of each component on the thermal properties of the pyrotechnic formulations and to determine the decomposition temperatures. The samples (25–30 mg) were heated from 20 °C to 550 °C, with 5 °C/min heating rates. The morphology of the pyrotechnic formulations was investigated with a VEGA II LMU scanning electron microscope (SEM), and a Bruker AXS energy-dispersive X-ray spectroscope (EDS—with QUANTAX 1.8.2 dedicated software) was employed to highlight the composition of the reaction products generated, indicated by the decomposition and oxidation equation of the SGD (12).

$$
\begin{aligned}
xKNO_3 + yP &+ zC_{26}H_{51}\backslash zC_{13}H_{12}O_2 \\
&\rightarrow \alpha K_2O + \beta N_2 + \gamma P_2O_5 + \delta P + \varepsilon C + \theta H_2 + \mu H_2O \\
&+\sigma CO_2 + \omega CO
\end{aligned}
\tag{12}
$$

The heat of combustion was evaluated using an AVL 1805 Ballistic instrument that includes a 25 cm$^3$ water calorimetric bomb set and a Beckman thermometer (0.01 °C precision). For this measurement $2 \pm 0.1$ g of each sample, that was previously grinded, was initiated in a closed vacuum bomb using a radiant wire with 1.5 cal/cm. The specific volume of the gases released from the calorimetric bomb was measured using a Julius Peters gas meter.

The combustion rate and combustion flow were studied using a HD SONY camera (digital display format 1920 × 1080 pixels, 1/2.8 CMOS detector and 30 × optical zoom) connected with a thermal sensor, TAU 2 Uncooled Cores 324 (*LWIR* thermal sensor-digital display format 324 × 256 pixels, uncooled detector, waveband 7.5–13.5 μm), placed 1.5 m from the combustion center. Ten measurements were performed for each sample and the average value was reported. Afterwards, by measuring the combustion duration, we calculated the combustion rate (13) and combustion flow (14) [25]. Thermal sensors measured the combustion temperature of the pyrotechnic composition. Meanwhile, the combustion coefficient (3) was also calculated to determine the aerosol quantity contained in the smoke cloud.

$$
v_c = \frac{\Delta H}{\Delta t_c}
\tag{13}
$$

$$
\dot{m} = \frac{\Delta m_{pc}}{\Delta t_c} = \rho \cdot S \cdot v_c
\tag{14}
$$

where $v_c$ is the combustion rate (m/s), $\dot{m}$ is combustion flow (g/s), $H$—cylindrical tablets height (mm), $m_{pc}$—charge mass (g) and $t_c$—combustion time (the moment between the combustion charge starts to when it ends).

The IR screening capacity was investigated using a constant volume, enclosed smoke testing chamber (cubic Plexiglas chamber with volume $V_{ch} = 0.512$ m$^3$), so external meteorological factors did not influence the results. In the IR sensor path, two rectangular openings were cut out of the smoke chamber and covered with plastic foil to allow the radiation emitted by the target to reach the thermal camera. The smoke chamber test configuration is represented in Figure S1 (from Supplementary Materials). The optical system described above for determining combustion rate and combustion flow was also used in this setup. Besides, an *MWIR* optoelectronic system (digital display format 640 × 512 pixels, cooling detectors generation III, optical zoom 20×, detection range 12 km, recognition range 4.5 km, wavebands 3–5 μm) was also used, for a complete IR analysis (Figure S4 from Supplementary Materials). Relative humidity during the testing was 45% and temperature was 25 °C.

Open field tests were designed to observe the target, measure the multispectral screening time, and include both a static and dynamic component. For these experiments, SGD with a diameter of 40 mm filled with 170 g of RP composition was used. For the static setup, four tests were performed that implied an SGD positioned at 80 m length from the sensor

and 3 m height (Figure S5 from Supplementary Materials). These tests were conducted at 22 °C, 45% average humidity and wind speed between 0.8–3 m/s.

For the dynamic setup, two tests were carried out that involved an SGD launcher that released a charge, which reached 10 m high with the same length from the sensor for static measurements (Figure S6 from Supplementary Materials). The images were analyzed with the same three types of sensors that were used in the static setup (HD SONY, *MWIR* optoelectronic system and *LWIR* thermal sensor TAU 2 Uncooled Cores 324) and the screening time (the time until the target becomes visible) was reported. These tests were conducted at 20 °C, 45% average humidity and wind speed between 1–3 m/s.

## 4. Results and Discussions

DTA measurements (Figure 3) allowed us to determine which of the compositions were more sensitive to temperature. PC1 presented the lowest self-ignition temperature at 329 °C, while a self-ignition temperature of 363 °C was determined for PC2 and 381 °C for PC3. Thus, the powder form of the composition with the lowest self-ignition temperature (PC1) was chosen because the most sensitive component was required to ignite the mixture faster and maintain the smoke in the first period after the device was launched, whereas tablets were shaped from the other two compositions. However, the self-ignition temperatures were relatively high, and they were raised when the mixture was pressed into a tablet shape, so the final smoke charge capsules required an initiation mixture to burn. Every sample had an endothermic transformation observed at 129 °C that was associated with $KNO_3$ crystal structure transformation [26]. Additionally, the melting point of paraffin was observed in PC2 and PC3 compositions between 50–60 °C [27].

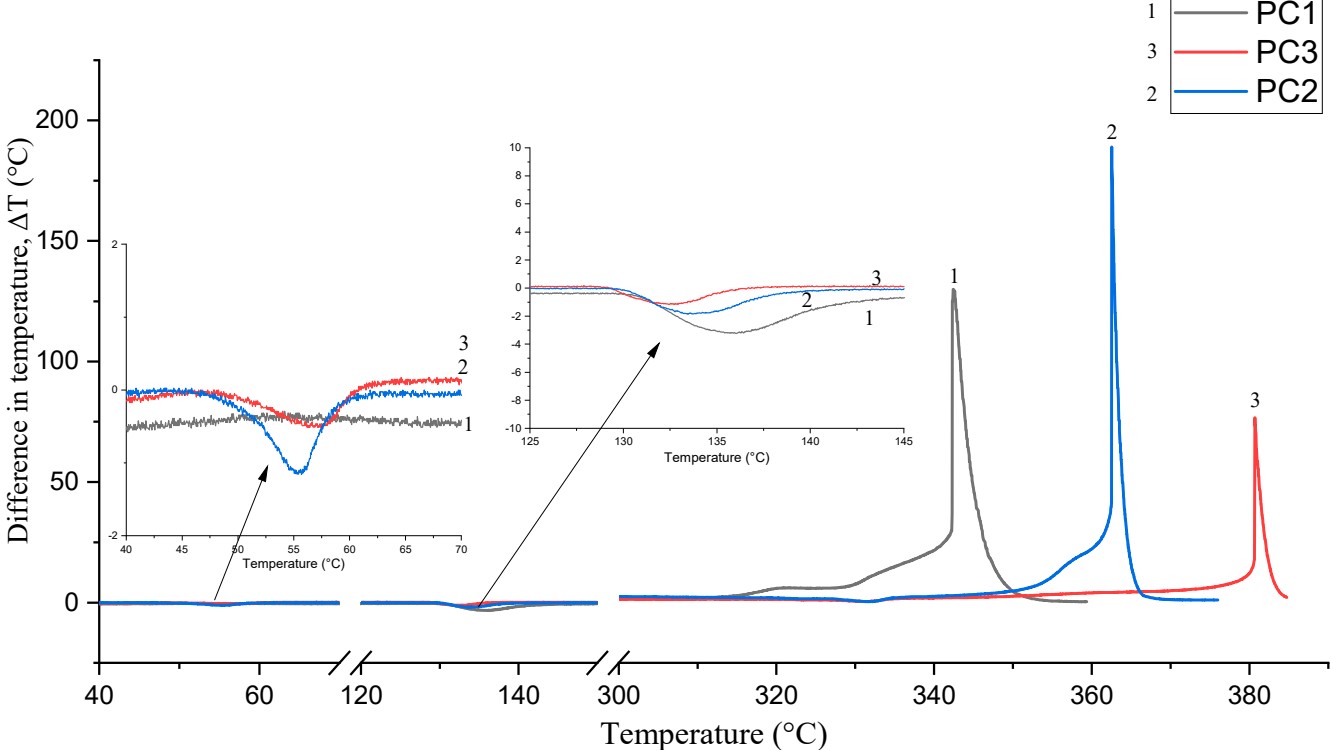

**Figure 3.** DTA thermogram of the smoke compositions.

Following the combustion of the composition, samples were examined with SEM/EDS to characterize the reaction products and visualize the morphology of the reaction products. According to SEM analyses, the resulting solid was distributed as conglomerations of two types of particles with random arrangement, irregular shapes, and dimensions ranging from 10–100 um. Some regions of the analyzed materials comprise phosphorus, magnesium, potassium, oxygen, and barium (Figure 4a") and have a porous form (Figure 4a), while some

particles contain only phosphorus, carbon and oxygen (Figure 4b"). All the resulting materials possess cracks in their structure (Figure 4a',b'). The samples taken from the bottom of the enclosed testing chamber were mostly porous particles, and the EDS elemental concentration profiles show that they mostly contain Mg and Ba, which were found in the ignition composition of the SGD (Figure S7 from Supplementary Materials). The samples taken from the chamber walls, on the other hand, show that the phosphorus was uniformly distributed throughout the enclosed smoke chamber volume and mostly formed spherical particles. The presence of phosphorus, barium, and oxygen as basic products of dispersed smoke was confirmed by EDS line profile measurements (Figure S8 from Supplementary Materials). The proportion of binder has no discernible effect on the thermodynamic properties of the mixture. The difference between compositions is a result of the quantity of oxidant used rather than the amount of red phosphorus. Compositions with a high phosphorus ratio, 35–50% (PC2 and PC3), provide a high percentage of gases, whereas compositions with a high oxidant ratio, up to 70%, provide a high combustion heat.

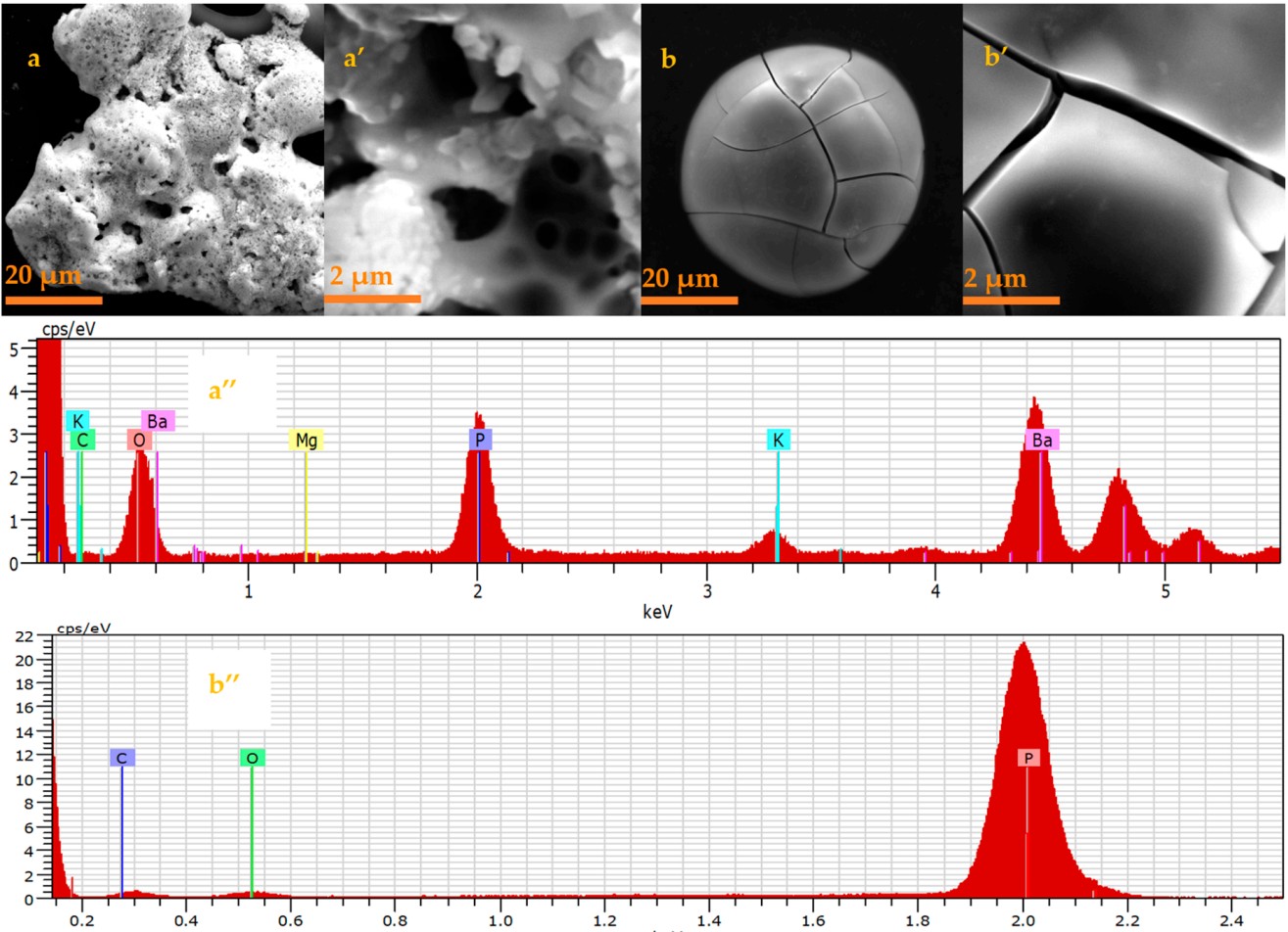

**Figure 4.** SEM images of resulting solid particle and the corresponding EDS spectra: (**a**)—porous shape of phosphorus and barium particle; (**a'**)—submicron cracks in the phosphorus and barium particle; (**b**)—spherical form of phosphorus and oxygen particle; (**b'**)—submicron cracks in the phosphorus and oxygen particle; (**a''**)—Phosphorus and barium particle; (**b''**)—Phosphorus and barium particle.

The determined heat of combustion and specific volume was: PC1—1165.9 kcal/kg and 209.9 L/kg, PC2—492.3 kcal/kg and 314.1 L/kg and PC3—647.7 kcal/kg and 343.3 L/kg.

However, PC3, which contains 50% RP, has a low combustion temperature compared to the other samples. The high RP content also influences the combustion rate, which has

the lowest value of 0.33 mm/s (Table 2), but offers a long combustion time and continuous smoke emission, up to 10 s. It was observed that increasing the proportion of potassium nitrate increases the combustion rate and temperature while decreasing the smoking time. The coefficient $k_c$ gives information about the quantity of aerosols generated by each type of composition, thus observing that the composition PC1 generates the highest quantity of aerosols after burning at the highest temperature of 1390 °C (Figure 5). As a result, not all of the tablet's mass contributes to the generation of aerosols. The residual elements are presented as too-large particles of ash that settle on the ground rather than forming an aerosol. The proportion of each component was optimized based on these properties to form a cloud with superior characteristics. Using three types of compositions in the smoke charge provides a combination of characteristics with optimal smoke cloud formation.

**Table 2.** The combustion rate and combustion flow for pyrotechnics sample.

| Sample | $m_{pc}$ (g) | $m_r$ (g) | $m_g$ (g) | $k_c$ (%) | H (mm) | D (mm) | $\rho$ (g/cm³) | $t_c$ (s) | $t_{VIS}$ (s) | $t_{IR}$ (s) | $v_c$ (mm/s) | $\dot{m}$ (g/s) | T (°C) |
|---|---|---|---|---|---|---|---|---|---|---|---|---|---|
| PC1 | 0.51 | 0.01 | 0.10 | 78 | 5.00 * | - | 1.06 | 3.04 | 3.66 | 3.41 | 1.65 | 0.17 | 1390 |
| PC2 | 0.57 | 0.02 | 0.17 | 67 | 3.55 | 11.71 | 1.49 | 4.22 | 5.15 | 4.52 | 0.85 | 0.14 | 1368 |
| PC3 | 0.58 | 0.01 | 0.19 | 66 | 3.10 | 11.74 | 1.62 | 9.55 | 11.05 | 10.05 | 0.33 | 0.06 | 1028 |

Where: $m_{pc}$—pyrotechnic composition mass (g); $m_r$—mass of the residue (g); $m_g$—mass of released gases (g); $k_c$—combustion coefficient (%); H—cylindrical charge high (mm); D—cylindrical charge diameter (mm); $\rho$—load density (g/cm³); $t_c$—combustion time of cylindrical charge (s); $t_{VIS}$—VIS smoke time (s); $t_{IR}$—IR smoke time (s); $v_c$—combustion rate (m/s); $\dot{m}$—combustion flow (g/s); T—combustion temperature; * in this situation H is represented by the length of the powder composition path.

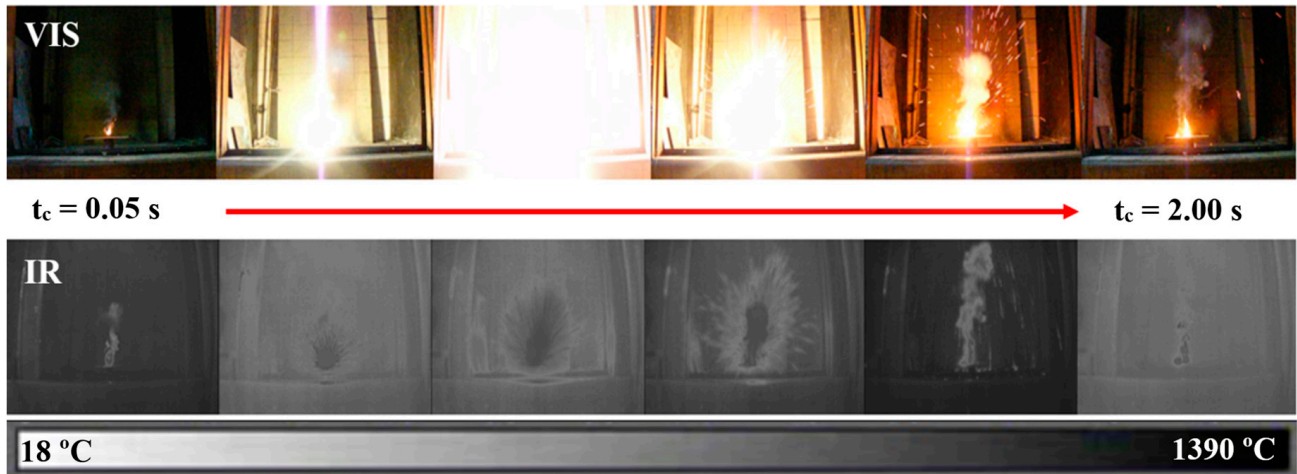

**Figure 5.** PC1 pyrotechnic tablets combustion in the first 2 s in VIS and IR mode.

The purpose of laboratory testing was to determine the efficacy of screening for small-sized samples of pyrotechnic mixtures. These tests were carried out under controlled and repeatable conditions and the results are presented in Table 3. It was possible to observe the evolution of the smoke over time and the radiation disturbance by keeping the amount of screening smoke generated in a known enclosed space constant. To illustrate how the test was performed, Figure 6 shows the images received for test number five at different times from the three optical sensors used. Even if after 7 s in *MWIR* (middle wavelength infrared) the objective can already be observed, in *LWIR* (long wavelength infrared) and in VIS it was not yet visible.

**Table 3.** Smoke chamber test results.

| Test No. | Sample | $m_{pc}$ (g) | $m_a$ (g) | $V_g$ (cm³) | $k_{oc}$ (%) | $\rho_a$ (g/cm³) | Screening Time *MWIR* (s) | Screening Time *LWIR* (s) |
|---|---|---|---|---|---|---|---|---|
| 1 | PC1 | 2.25 | 1.89 | 470.25 | 0.092 | 3.69 | 3.63 | 5.12 |
| 2 | PC1 | 5 | 4.2 | 1045 | 0.204 | 8.23 | 6.21 | 8.19 |
| 3 | PC1 | 10 | 8.4 | 2090 | 0.480 | 16.40 | 11.22 | 14.32 |
| 4 | PC1 | 15 | 12.6 | 3135 | 0.612 | 24.6 | 16.01 | 19.21 |
| 5 | 2 × PC2/2 × PC3 | 1.14 1.08 | 1.45 | 762.1 | 0.149 | 2.83 | 7.12 | 8.19 |
| 6 | 4 × PC2/4 × PC3 | 2.28 2.16 | 2.91 | 1554 | 0.304 | 5.68 | 13.86 | 16.12 |
| 7 | 8 × PC2/8 × PC3 | 4.56 4.32 | 5.82 | 3154 | 0.616 | 11.36 | 18.93 | 21.84 |
| 8 | 12 × PC2/12 × PC3 | 6.84 6.48 | 8.74 | 4399 | 0.850 | 17.07 | 23.05 | 25.45 |

Where: $m_{pc}$—pyrotechnic composition mass (g); $m_a$—mass of aerosols (g); $V_g$—gas volume (cm³); $k_{oc}$—coverage coefficient parameter (%); $\rho_a$—aerosol density (g/cm³). Note: smoke curtain generated in a constant volume kept the screening effect throughout the test until the smoke was released from the enclosure (after t > 240 s) for VIS.

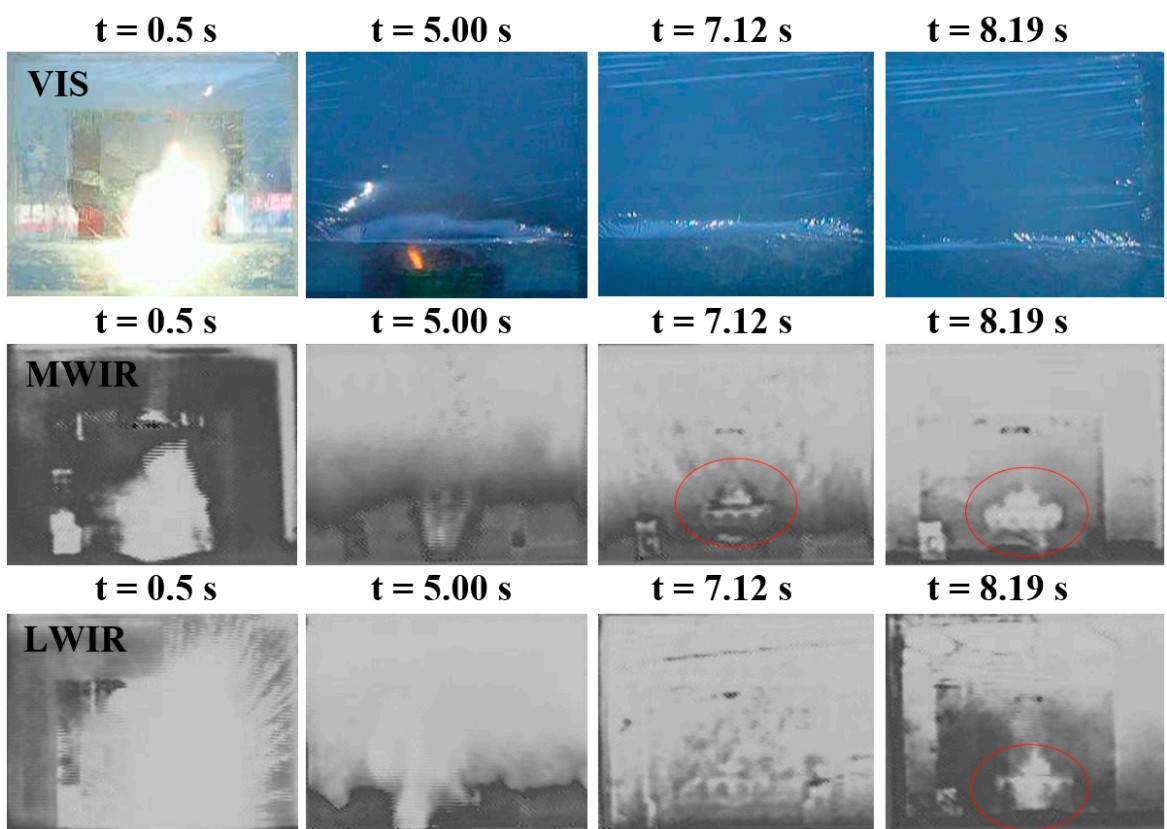

**Figure 6.** VIS, *MWIR* and *LWIR* imaging.

IR screening time can vary slightly depending on weather conditions (low temperature, humidity > 50%, sunny or cloudy day) and contrast between target and background. These values were measured in optimal conditions as we maintained a high contrast and fixed volume inside the enclosure. Experimental values of concentration and screening time highlighted the fact that pyrotechnic mixtures based on red phosphorus have a higher screening efficiency in VIS, the screening time being given by the stability of the smokescreens dispersed in the environment (depending on the wind speed and direction). The maximum VIS screening time can only be determined under field conditions for a wind speed value less than 3 m/s [28–30]. The screening effect occurs in VIS from the

moment the smoke curtain was generated, and the area of interest was covered, whereas the screening effect occurred in IR from the moment the pyrotechnic mixture was ignited. The main goal of the smoke chamber test was to keep the aerosol concentration uniform for an extended period, allowing the temperature influence of the dispersed particles on the screening process to be observed. The screening goal will be met if a higher concentration of aerosol maintains a temperature above the target temperature. As a result, we can mention that the IR screening time is inversely proportional to the cooling time of the particles below the target temperature.

The screening times obtained demonstrate the effect of each mixture's thermodynamic parameters as well as the loading form. The difference in screening times (ST) obtained for *MWIR* and *LWIR* was due to the different performance characteristics of the two systems working in different spectra. Thermal equipment with a cooling system that operates between 3–5 μm outperforms equipment without a cooling system that operates between 8–14 μm. To write the screening time equation in IR, only the results obtained for the tests five to eight from Table 3 (PC2–PC3) were considered, because the variation of their parameters as a function of time remained constant. Finding the composition of PC1 (12% of the full charge), having the role of a rapid smoke generator of the cloud and the shortest screening time in IR compared to the PC2–PC3 mixture (88% of the full charge), was not considered necessary in the screening time calculation.

To adjust the relation (10) with the experimental IR smoke time value it was necessary to introduce into the equation an empirically determined value: the proportionality coefficient $f_a(\lambda) = 0.775 \cdot 10^{-\gamma}$ (s·m$^2$/g) for *LWIR* and $f_a(\lambda) = 0.875 \cdot 10^{-\gamma}$ (s·m$^2$/g) for *MWIR*. This coefficient is specific to RP-based compositions and depends on the screen cloud volume ($V$), where $\gamma$ values are the following: $\gamma = 0$ for $V < 1$ m$^3$, $\gamma = 4$ for $V < 100$ m$^3$, $\gamma = 3$ for $V > 100$ m$^3$ and $\gamma = 2$ for $V > 1000$ m$^3$.

Static and dynamic tests in various atmospheric conditions were used to determine the physical parameters of the screening cloud generated by a single SGD. The experimental screening time was measured under the given conditions, and the theoretical value was calculated based on the stated relation (11) and the established physical parameters, allowing the results to be compared and the error level to be ascertained. Table 4 contains these values as well as information about the testing conditions. To exemplify how the test was performed, Figure S9 (from Supplementary Materials) shows the images received for test number three at different times from the LWR optical sensors. The variety of values of screening times in IR and VIS was given by the influence of the meteorological parameters, which were constantly changing, and other natural factors in the atmosphere. Therefore, initially, the capacity of screening smoke was analyzed at a constant volume. The variation in screening time and covering surface was inversely proportional to the wind speed and SGD ignition point. As a result, the stated calculation relation yields the empirical screening time with a probability of trust greater than 50%, as shown in Table 4, with high credibility (98%) for wind velocities lower than 1 m/s. This probability was considered with a high level of confidence, more significant than that observation probability obtained by the Jonson Equation (1) (which gives maximum probabilities of 50% [21]). The calculated screening time tends to be longer in the case of static tests (with a small coverage area) and shorter in the case of dynamic shootings (where the coverage area was much larger). According to the theoretical time values, a significantly larger scattering of the smoke cloud (due to the SGD ignition point) has a negative influence on the screening time, with the same mass of aerosols but in a much larger dispersed volume. In experimental testing, the measured screening times were influenced by wind speed. This parameter cannot be considered in the theoretical calculation and was a parameter that can significantly influence the difference between the two values (ST in static and dynamic testing). It was observed that the smallest error was obtained in the case the second set-up, in which the recorded wind speed was the lowest value ($v_v = 0.8$ m/s). The difference between the screening times *MWIR* and *LWIR* domains was also obtained during open field experiments, being

given by the differences in performance between the observation systems, which operate at different wavelengths.

**Table 4.** Results of experimental tests in field conditions.

| No. | Set-Up | Wave-Length | $n$ | $t_d$ (s) | $t_m$ (s) | ST | Error (%) | P (%) | Target (cm) | $S_t$ (LsxHs) (m²) | $l_s$ (m) | $V_s$ (m³) | $S_a$ (m²) | $k_s$ (%) | $v_v$ (m/s) |
|---|---|---|---|---|---|---|---|---|---|---|---|---|---|---|---|
| 1 | Static | VIS MWIR LWIR | 1 | 1.16 | 7.51 1.38 1.68 | N/A 2.06 2.32 | N/A 49 38 | N/A 51 62 | Person 60 × 180 | 5.6 × 4.7 | 5 | 131 | 4.3 × 3.5 | 37.61 | 2.8 |
| 2 | Static | VIS MWIR LWIR | 1 | 1.33 | 26.43 2.07 2.68 | N/A 2.44 2.75 | N/A 17 2 | N/A 83 98 | Person 60 × 180 | 7.2 × 4.1 | 5 | 147 | 3.6 × 3.4 | 51.20 | 0.8 |
| 3 | Static | VIS MWIR LWIR | 1 | 1.12 | 14.23 1.48 1.86 | N/A 2.23 2.52 | N/A 50 35 | N/A 50 65 | Car 400 × 185 | 5.9 × 4.6 | 5 | 135 | 3.6 × 3.2 | 42.41 | 1.6 |
| 4 | Static | VIS MWIR LWIR | 1 | 1.21 | 12.4 1.42 1.82 | N/A 1.94 2.19 | N/A 36 20 | N/A 64 80 | Car 400 × 185 | 6.2 × 4.8 | 5 | 148 | 3.8 × 3.2 | 40.90 | 1.8 |
| 5 | Dynamic | VIS MWIR LWIR | 1 | 1.12 | 14.1 1.38 2.12 | N/A 1.20 1.35 | N/A 13 36 | N/A 87 64 | Panel 200 × 200 | 19.8 × 22.2 | 7 | 3077 | 8.2 × 19.7 | 36.75 | 1.2 |
| 6 | Dynamic | VIS MWIR LWIR | 2 | 1.16 | 25.3 2.16 3.05 | N/A 1.79 2.03 | N/A 17 33 | N/A 83 67 | Panel 200 × 200 | 27.6 × 24.3 | 7 | 4694 | 14.4 × 19.5 | 41.86 | 1.4 |

Where: $n$—SGD number; $t_d$—dissemination time (s); $t_m$—smoke time (s); $ST$—screening time; Error—error level between experimental and theoretical IR screening time value (%); P—trust probability of ST, P = 100-Error (%); $S_t$—cloud total surface (m²); $S_a$—cloud active surface (m²); $l_s$—cloud width (m); $V_s$—smoke volume (m³); $k_s$—active smoke coefficient (%) $v_v$—wind velocity (m/s).

For adjusting the mathematical model (1–11) for a complete SGD the following constant parameters were used: $V_{sp}$ = 288 (L/kg), $k_c$ = 0.71, $k_s$ = 0.479, $m_{pc}$ = 170 g, $m_a$ = 120.7 g, $V_g$ = 0.049 m³, $V_s$ = 3077 m³, $k_{oc}$ = 0.002, $l_s$ = 5 m and $CS$ = 9.395 g/m². At the same time, the multiplier number, n, was introduced (for the use of more than one SGD). The average value of the scattering factor of 47.9% was obtained from the analysis of the physical parameters determined by real measurements. Under these conditions the Equation (11) becomes:

$$ST_{LWIR} = 9.395 \cdot \frac{R}{d_c} \cdot \frac{1}{\nu(\lambda)} \cdot 0.875 \cdot 10^{-\gamma} \tag{15}$$

$$ST_{MWIR} = 9.395 \cdot \frac{R}{d_c} \cdot \frac{1}{\nu(\lambda)} \cdot 0.775 \cdot 10^{-\gamma} \tag{16}$$

The coverage area of interest was the area at the base of the smoke, making the dispersion of the cloud from low heights more efficient (the dispersed volume of smoke being smaller but with a high density of aerosols). However, the tactical requirements of combat involve the dispersion of a smoke cloud ten times the size of the target in order to cover a military vehicle, for instance.

## 5. Conclusions

This research describes the design and evaluation of screening smoke compositions based on red phosphorus, in laboratory conditions and also in open field conditions. This work proposed a mathematical approach to explain the screening time of an objective by optimizing the smoke compositions, thus reducing the amount of mass required to provide an acceptable screening time.

Three types of red phosphorus mixtures were used to create an optimal smoke composition that was further tested. First, the temperature sensitivity of the compositions was assessed using DTA thermal testing and the morphology of the combustion products was also studied using SEM/EDX.

The main physiochemical properties for obtaining the screening time mathematical model were established through laboratory investigation that included smoke chamber and open field testing.

Although red phosphorus-based pyrotechnic compositions offer IR screening properties, obtaining a long screening time requires the use of numerous SGD. Typically, a vast amount of pyrotechnic formulation/mixture was utilized to produce a high concentration of aerosols in the atmosphere.

The experiments' positive results ultimately resulted in the creation of new equations that expand the possibilities for determining the performance of pyrotechnic screening systems in the visible and infrared spectra while decreasing the RP composition quantity. The pyrotechnic screening effect can be theoretically defined by determining the minimum aerosol concentration and the screening time in IR using the defined mathematical model.

The same types of testing can be performed for various kinds of smoke compositions in future research. This will allow the development of a universal calculation relationship that can be applied to any IR smoke-generating device. With the help of such a universal correlation, software that can simulate the final effect on any type of situation can be created. Thus, the production costs can be significantly reduced by simplifying the steps dedicated for testing, evaluation and design of the optimal composition that creates the desired effect requested by the final user.

**Supplementary Materials:** The following supporting information can be downloaded: https://www.mdpi.com/article/10.3390/app122412893/s1. Figure S1: Visual exemplification of Experimental determinate of smoke volume in open field condition; Figure S2: Display of the SGD that were used in the investigations; Figure S3: Exemplification in visual form for experimental measurement of smoke volume in open field conditions; Figure S4: Smoke chamber test configuration and visual form of experiment; Figure S5: Static firing setup; Figure S6: Dynamic firing setup; Figure S7: SEM/EDS mapping and line profile of the base of the enclosed testing chamber; Figure S8: SEM/EDS mapping and line profile of the walls of the enclosed testing chamber; Figure S9: Exemplification in visual form for experimental measurement.

**Author Contributions:** Conceptualization and methodology, B.G.P. and D.P.; Investigation B.G.P., D.P. and E.T.; Writing-original draft preparation, D.P., E.T., G.T. and F.M.D.; Validation, T.R.; Data curation and SEM investigation, T.R., R.E.G. and F.M.D. All authors have read and agreed to the published version of the manuscript.

**Funding:** This research received no external funding.

**Institutional Review Board Statement:** Not applied.

**Informed Consent Statement:** Not applied.

**Data Availability Statement:** All data generated or analyzed during this study are included in this published article (and its Supplementary Information Files).

**Acknowledgments:** The authors want to acknowledge PRO OPTICA S.A, Bucharest, company for providing all the optical systems used for investigations.

**Conflicts of Interest:** The authors declare no conflict of interest.

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
