# Peer review of "Design and Evaluation of Screening Smoke Compositions Based on Red Phosphorus in Open Field Conditions"

_applsci, doi:10.3390/app122412893_

Round 1
Reviewer 1 Report
The article describes the design and evaluation of screening smoke compositions based on red phosphorus. The use of such a composition is an ecological solution and is worth developing. The paper is generally well written but still contains several typos here and there.
1 Check and correct the notation of chemical formulas, e.g. KNO3.
2. The notation of units is wrong in some places. We only write % without spaces.
3. Combustion rate and combustion flow - bad formula reference in the text or bad formula numbering.
4. Notation of counts - once with a comma, once with a period. Please standardize.
5. A magnesium-based ignition composition - what is behind such a statement, what mixture?
6. In what form was black powder used?
7. Table 1 shows the composition of the primary components. For PC1, the total is 92%. What is the 8%?
8. Some units are missing superscripts, such as m3.
9. In Table 3, the designations 2xPC2 2xPC3 and other similar designations appear. What do these mean? Please explain the exact compositions for these samples.
Author Response
We are very grateful for the reviewer point of view for our manuscript. The comments are encouraging, and the reviewer appear to share our judgement that this study and its results are important. Please see below, in red, our detailed response to comments.
Point 1: Check and correct the notation of chemical formulas, e.g. KNO3.
Response 1: Thank you for noticing the incorrect formulations. We did another proofreading, and we corrected all the errors that we found.
Point 2: The notation of units is wrong in some places. We only write % without spaces.
Response 2: We found the mistake in Table 1 and we correct it. Also, we checked the notation of the units in the manuscript, and we change what was flawed.
Point 3: Combustion rate and combustion flow - bad formula reference in the text or bad formula numbering.
Response 3: Thanks for your recommendations. Consequently, we adjusted the equations numbering.
Point 4: Notation of counts - once with a comma, once with a period. Please standardize.
Response 4: We are thankful to you for seeing this mistake, and we fix it by replace all the data that contained a comma, with a period.
Point 5: A magnesium-based ignition composition - what is behind such a statement, what mixture?
Response 5: Usually, for the purpose of ignite the thermal low-sensitivity compositions we use a standardized starter mixture named RR4 form UM Băbeni SA. The mixture is based on magnesium, but it also contains barium nitrate – Ba(NO3)2, barium peroxide – BaO2 and iditol. We updated the materials list to also include the reference to this mixture.
Point 6: In what form was black powder used?
Response 6: Prior to use, the gunpowder was sieved to obtain particles with a size of less than 1 mm. The gunpowder was in a pulverulent form. We update the materials list to include the particle size of the black powder.
Point 7: Table 1 shows the composition of the primary components. For PC1, the total is 92%. What is the 8%?
Response 7: We sincerely regret this error. We failed to see that the phosphorus value in the table was incorrect. Instead of 20, as the table states, this is actually 28. The required adjustments were made to table 1.
Point 8: Some units are missing superscripts, such as m3.
Response 8: In that context m3 was, in fact, the measure of the volume and needed to pe written as m3. We are apologising for the misleading and we change it in the text.
Point 9: In Table 3, the designations 2xPC2 2xPC3 and other similar designations appear. What do these mean? Please explain the exact compositions for these samples.
Response 9: The composition and the form of the PC2 and PC3 is described in section `3.1.2. Preparation of the Pyrotechnic Compositions`, where is stated that: PC2 and PC3 were compressed into cylindrical shapes (θ = 11 mm and h=3 mm) followed by drying. The samples that are expressed as 2xPC2 2xPC3 and similar, are in fact two PC2 tables along with two PC3 tablets, and so on. The tablets were placed in a cylindrical container (the size and shape of a smoke generating device), in the form of a pile, exactly as it would simulate their position in an SGD.
Reviewer 2 Report
This is a very good paper in the finest tradition of Rotariu et al. reporting on the design and evaluation of red P based smoke formulations. The paper is very suitable for publication in Applied Sciences.
My comments are minor:
(i) p 3, after eq. (4): according to IUPAC lieter should be capitalized "L" and not "l".
(ii) p 5, Tab. 1: composition PC1 does not add up to 100%.
(iii) On the bottom of p 6 is eq (12) and in the center of p 7 is again eq. (12).
Please re-number.
(iv) Ref. 17: Names of authors are incorrect and also some authors are missing. Also, please provide volume and page numbers.
(v) Ref. 32: Names of authors are incorrect and also some authors are missing. Also, please provide volume and page numbers.
Author Response
We are very grateful for the reviewer point of view for our manuscript. The comments are encouraging, and the reviewer appear to share our judgement that this study and its results are important. Please see below, in red, our detailed response to comments.
Point 1: p 3, after eq. (4): according to IUPAC lieter should be capitalized "L" and not "l".
Response 1: Thank you for your reference. We made the modification in the entire content of our paper where this mistake appears.
Point 2: p 5, Tab. 1: composition PC1 does not add up to 100%.
Response 2: We sincerely regret this error. We failed to see that the phosphorus value in the table was incorrect. Instead of 20, as the table states, this is actually 28. The required adjustments were made to table 1.
Point 3: On the bottom of p 6 is eq (12) and in the center of p 7 is again eq. (12). Please re-number.
Response 3: Thanks for your recommendations. Consequently, we adjusted the equations numbering.
Point 4: Ref. 17: Names of authors are incorrect and also some authors are missing. Also, please provide volume and page numbers.
Response 4: Thanks for your suggestions. We revised all the references in the text and corrected the inaccurate ones.
Point 5: Ref. 32: Names of authors are incorrect and also some authors are missing. Also, please provide volume and page numbers.
Response 5: Thanks for your suggestions. We revised all the references in the text and corrected the inaccurate ones.
Round 2
Reviewer 1 Report
Thank you for your corrections.